# Pantalar Intact Dislocation: A Systematic Review

**DOI:** 10.3390/jfmk10010055

**Published:** 2025-02-03

**Authors:** Eleonora Dell’Agli, Marco Sapienza, Mirko Domenico Castiglione, Maria Agata Musumeci, Sebiano Pitronaci, Andrea Sodano, Vito Pavone, Gianluca Testa

**Affiliations:** Department of General Surgery and Medical Surgical Specialties, Section of Orthopedics and Traumatology, A.O.U. Policlinico-San Marco, University of Catania, Via Santa Sofia 78, 95123 Catania, Italy; dellagli.eleonora@hotmail.it (E.D.); mirkodomenicocastiglione@gmail.com (M.D.C.); musumeci.maria.mm@gmail.com (M.A.M.); sebianopitronaci@gmail.com (S.P.); asodano97@gmail.com (A.S.); vitopavone@hotmail.com (V.P.); gianpavel@hotmail.com (G.T.)

**Keywords:** talar dislocation, talar injury, clinical presentation, pantalar dislocation, outcomes

## Abstract

Background: This scoping review analyzes the available literature on pure total talar dislocation, focusing on its epidemiology, clinical presentation, imaging techniques, surgical options, rehabilitation protocols, and complications. Methods: Following the PRISMA-ScR guidelines, a comprehensive search was conducted across the PubMed, Web of Science, and Scopus databases. The search yielded 185 articles, of which 30 satisfied the inclusion criteria and focused on pure total talar dislocation without fractures. Data from each study were extracted, including patient demographics, injury characteristics, treatment methods, and outcomes. Results: The studies included case reports, case series, and reviews. Despite the heterogeneity of the studies, the key findings suggest that early reduction, careful wound management, and soft tissue preservation are crucial in minimizing complications such as avascular necrosis (AVN), post-traumatic arthritis, and infection. The long-term outcomes varied, and the risk of AVN remained high, particularly in cases with compromised blood supply to the talus. Conclusions: Pure total talar dislocation is a rare and challenging condition with no established management protocol. While talar reimplantation and joint fixation offer promising outcomes in preserving function, the risk of complications, particularly AVN, remains significant. Additional research is necessary to standardize treatment protocols and improve clinical outcomes for this rare but severe injury.

## 1. Introduction

Pantalar dislocation without an associated fracture, defined also as pure total talar dislocation, is a rare ankle lesion that accounts for 0.06% of all dislocations and only 2% of all talar injuries [1]. In 1680, Fabricius von Hildanus, the “father of German surgery”, reported the first case of total talar dislocation without concomitant fracture [2].

Ankle injuries involving the talus can be severe. High-energy traumas can cause complete talus dislocation, affecting multiple joints. The talus, lacking muscular attachments, is prone to dislocation [2]. This injury typically results from the ankle pointing downward and twisting outward [3].

Pantalar dislocation is usually an open injury involving nearby ligaments, capsular attachments, and bones (malleoli, navicular, or calcaneus). It also often leads to soft tissue damage, wound contamination, and disruption of the talar blood supply [4]. In particular, the missing talus is defined as a completely extruded talus lesion without remaining soft tissue attachments [5,6].

However, information about pantalar dislocation is rare in the orthopedic literature, which makes it challenging for surgeons to fully understand and treat this condition. While there is no standard protocol to treat pantalar dislocation, prompt treatment can reduce complications and improve outcomes [4]. To preserve the blood supply of the talus, an early closed reduction is performed. If this procedure fails, an open reduction is carried out [7,8].

Talar dislocations can be categorized based on the involvement of different joints. They can be classified as mono-articular dislocations (i.e., occurring within the talocrural joint), bi-articular dislocations (i.e., affecting both the talocalcaneal and talonavicular joints), or tri-articular dislocations (i.e., involving the talocrural, talocalcaneal, and talonavicular joints), which are also known as luxatio tali totalis [9,10,11,12].

Different methods can be used to stabilize the joint, depending on the surgeon’s experience. In cases of a missing talus, reattaching the native talus after thorough irrigation and debridement is recommended, despite the increased risk of complications such as avascular necrosis (AVN), infection, and post-traumatic arthritis; such complications may require secondary surgeries such primary talectomy and tibio-calcaneal arthrodesis [6,8].

This review analyzes the available literature related to pure total talar dislocation, highlighting its epidemiological data, clinical and imaging characteristics, management, treatment, and outcomes.

## 2. Materials and Methods

### 2.1. Search Selection

This systematic review was conducted according to the guidelines of the Preferred Reporting Items for Systematic Reviews and Meta-Analyses (PRISMA). This systematic review was registered in the International Prospective Register of Systematic Reviews (PROSPERO: CRD42024615209).

For detailed information regarding the PRISMA checklist, refer to the Appendix A (PRISMA checklist).

Our goal was to summarize the primary features of pure pantalar dislocation and analyze available studies in the literature; we excluded articles that focused on associated talar fractures. The presence of talus osteochondral defects did not represent an exclusion criterion.

We examined several electronic databases (PubMed, Web of Science, and Scopus) in September 2023. The research was conducted from January through June 2023, and the keywords we used included ((“pantalar dislocation” OR “pantalar dislocations”) OR (“total talar dislocation” OR “total talar dislocations” OR “total talar extrusion” OR “total talus dislocation”)) NOT (fracture). MesSH terms were included. We did not impose any temporal limitations about when the studies were published. A total of 185 articles were retrieved. Studies providing any level of evidence about pure total talar dislocation were considered eligible for this study (Appendix B).

### 2.2. Data Extraction

The main parameters of each article were tabulated; we extracted data pertaining to epidemiology, clinical and radiological presentation, treatment, rehabilitation, and complications.

### 2.3. Quality Assessment

Five authors (M.D.C., E.D., M.A.M., S.P., and A.S.) individually read the selected articles and evaluated and discussed their quality. M.D.C., E.D., and M.A.M. identified the research at the level of titles, abstracts, and full texts. S.P. and A.S. extracted the data. The senior author (V.P.) intervened in ambiguous cases.

## 3. Results

### 3.1. Search Results

Of the 185 studies initially selected, 101 were excluded because they were reported more than once. After the first preliminary screening, 33 additional articles were removed because their titles or abstracts were not consistent with the main topic.

Twenty-one studies were also excluded: twelve because the full text was not available, two because they were not written in English, and seven because they were not related to the main topic analyzed. Therefore, 30 studies were included in the final sample. All authors collectively re-evaluated their choices to ensure accuracy. The PRISMA flowchart is shown in Figure 1.

The 30 articles analyzed were mostly case reports and case series consisting of small numbers of patients; pantalar dislocation is a very rare injury.

A total of 108 patients were included in our review. The predominant gender was male (*n* = 74). The mean age of the cohort was 34 years (range: 17–51 years). High-energy trauma was experienced by 60 of the patients that yielded an open lesion of the ankle that required accurate irrigation and debridement. Treatments varied, and an open reduction and stable ankle joint without any devices was noted in 31 cases. The treatment in the remaining cases consisted of a single or multiple K-wire (*n* = 14), external fixation (*n* = 30), K-wire and external fixation (*n* = 10), screws (*n* = 4), Steinmann pin (*n* = 3), and Steinmann pin and external fixation (*n* = 9). Only in one case was amputation necessary.

There were six cases of a missing talus and severe contamination that led the surgeons to initially use a cement spacer supported by the external fixation to maintain the congruency of the ankle joint. Once the infection and the condition of the soft tissues were controlled in three cases, a prosthesis was implanted. In three cases, ankle fusion was performed with reimplantation of the native frozen talus and arthrodesis. After the treatment, the mean follow-up time was 30 months (range: 2–108 months). Complications were noted in 62 patients: infection (*n* = 10), AVN (*n* = 32), clinical osteoarthritis (*n* = 16), and pain (*n* = 4). In eight cases, these complications required arthrodesis.

### 3.2. Basic Characteristics of the Included Studies

We examined each study’s reference list. The selected articles are summarized in Table 1.

## 4. Discussion

Total talar dislocation is defined as the dislocation of the talus from the tibiotalar, talocalcaneal, and talonavicular joints; it is also referred to as “floating foot” [38].

Total talar dislocation accounts for only 3.4% of talar injuries; total extrusion without concomitant fracture is even rarer and accounts for only 0.06% of all dislocations and 2% of talar injuries [39]. However, it appears that, as trauma kinetics have changed over time, total dislocation is becoming more frequent [40].

The typical mechanism of talus dislocation involves a combination of plantar flexion with either supination or pronation of the foot [41]. Despite the lack of muscular or tendinous attachments on the talus, which theoretically increases its own vulnerability to dislocation, this injury is rare due to the talus’s deep position within the tibio-peroneal mortise [42].

The talus’s position is primarily maintained by its ligamentous support, including the tibiotalar deltoid ligaments medially, the anterior/posterior talofibular ligaments laterally, and the interosseous ligament [1].

Talar dislocations can be categorized as either subtalar or pantalar. Subtalar dislocations involve the talocalcaneal and talonavicular joints, and pantalar dislocations additionally include the tibiotalar joint. Pantalar dislocations can present as anterolateral (most common), posteromedial, or posterior. Despite the frequency of open and anterolateral talar dislocations (more than 50% of cases) [43], anteromedial and posteromedial dislocations are less common; the latter are extremely rare [44].

In cases of extrusion, the talus may either remain attached via capsular attachments or be totally expelled through the skin and absent at the time of initial presentation [15]. Such high-energy trauma is often associated with events such as falls from a great height or motor vehicle accidents [39].

Leitner described a three-stage mechanism for anterolateral dislocation, considering talar extrusion as the final stage of combined subtalar supination and tibiotalar plantar flexion, which pulls the talus out of the ankle mortise [3]. Talar dislocation typically occurs in axial trauma, where plantar flexion combines with excessive inversion or eversion of the subtalar joint, leading to lateral dislocation in inversion cases or medial dislocation in eversion cases.

In a typical presentation, a fall with the ankle in a pointed-down and inward position can cause injury to the heel bone. The twisting of the joint in the foot can lead to the dislocation of the talus bone and damage to the ankle’s inner structures. The force of the injury can also harm the front and side capsules of the ankle, the ligament connecting the tibia and fibula in the front, and the membrane between the two lower leg bones. The absence of a fibula fracture could be due to the dissipation of energy across two main injury points: the talus dislocation and ankle injury [35].

### 4.1. Clinical Presentation

Upon arrival at the emergency department, it is evident from the initial physical examination that a patient has a pantalar dislocation. The patient is unable to walk on the injured foot and complains of severe pain. In most cases, the extreme forces that caused the dislocation would result in the talar extruding through an open wound [1]. The patient’s skin appears to be lacerated more often along the lateral or anterolateral aspect of the ankle and the talus is partially extruded from its articulation; it is sometimes held to the hindfoot simply by a few soft tissues or is even completely enucleated. This rare and peculiar injury is known as “missing talus” [5,26,28].

In the case of motor vehicle accidents, the extruded talus may be found hours later at the scene of the accident [23].

The wound, subcutaneous tissues, and the bone itself may be contaminated and at high risk of infection [28,37].

In less frequent closed dislocations, the lower extremity is visibly deformed. The ankle joint and foot are swollen and the skin is tight due to the prominence of the underlying talar head [45]. Superficial abrasions or even blisters can often be observed [20,24]. The foot is usually plantarflexed and pronounced with lateral displacement or supinated with medial displacement according to the dislocation mechanism and direction [46].

The neurovascular status of the extremity can be compromised: the tibialis posterior and dorsalis pedis pulses are not always palpable, and possible motor or sensory deficits can be present. When impaired, neurovascular status can be restored after the dislocation reduction [22].

### 4.2. Imaging

The diagnosis of pantalar dislocation is typically confirmed via an X-ray evaluation [1].

A three-dimensional (3D) Computed Tomography (CT) scan is recommended to check for additional fractures or injuries not visible on X-ray imaging. CT images of the original talus or 3D mirroring of the opposite talus are often used as references for creating prostheses [37].

Avascular necrosis, which typically appears six months to two years after the injury, is one of the most worrisome consequences [43]. Imaging can help identify early signs of revascularization; the Hawkins sign is the only early predictor on conventional radiographs. The Hawkins sign is observed six to eight weeks after injury as a subchondral radiolucency in the talar dome; it indicates early subchondral atrophy [47]. However, Magnetic Resonance Imaging (MRI) continues to be the most specific and sensitive technique for detecting early AVN development in the postoperative phase, and it is essential to allow the patient to bear weight on the affected foot [43].

### 4.3. Surgical Techniques

Total talar dislocation is rare, and the best approach for managing complete talus dislocation remains controversial [48]. However, generally, the basic principles of correct management for talus injuries include early reduction, joint fixation if unstable, careful soft tissue handling, and thorough wound care in cases of open dislocation [18,48].

Historically, primary talectomy and tibio-calcaneal arthrodesis were recommended to manage pantalar dislocation and reduce complications [24]. However, primary talectomy leads to leg-length discrepancies and hinders hindfoot functionality, limiting future reconstructive options [6,17,33]. The recent literature suggests that the accurate removal of damaged tissue, reinsertion of the talus, and temporary fixation of the affected joints is associated with satisfactory clinical outcomes; this process preserves bone stock and restores talar height [22,28,33].

Furthermore, recent case series of open pantalar dislocation have proven that risks of osteomyelitis and AVN are intimately associated with time to reimplantation; the best prognosis is associated with reimplantation performed within three hours. The poorest prognosis is linked to reimplantation performed over 24 h later, and secondary tibio-calcaneal arthrodesis can become essential [5,18,34].

In general, most researchers suggest restricted use of primary talectomy and tibio-calcaneal arthrodesis for cases of severe and gross contamination not liable for a satisfactory debridement and likely evolving to complications, or when the time of reimplantation is delayed [5,17,22,34]. The same approach is generally considered in the face of complications such as infection, early post-traumatic arthrosis, or AVN [5,17,22].

Information from the recent literature can help orthopedic surgeons faced with pantalar dislocation. Firstly, in cases of closed pantalar dislocation, an early closed reduction maneuver, first described by Mitchell, is warranted to prevent pressure necrosis of the overlying skin [24,25,36,48]; nevertheless, most authors suggest only a few attempts because of possible detrimental side effects on talar vascularity. If closed reduction is attainable, the results are typically good [6,7]; however, closed reduction is generally not achievable [1,28], mainly due to the trapping of the talar neck between the flexor tendons, buttonholing of the head of the talus between the tendons of the tibialis posterior and flexor digitorum longus, and nondisplaced medial malleolus [6]. If all attempts at closed reduction fail, open reduction must be performed. This procedure typically exploits an antero-lateral approach to that talus and eventually combines a medial approach that allows for a complete view of the dislocated joint or engaged talus [6,7,48,49]. This management can be largely applied, even for open dislocation, using the wound as a surgical approach. In that case, researchers suggest a single surgical debridement with primary closure to minimize the risk of infection [28].

Debridement is an important step in cases of open dislocations because it reduces the risk of infection [28]. Generous irrigation with sterile normal saline [28] and antibiotics, such as cefazoline or gentamicin/clindamycin or 5–10% povidone iodine [8,17,18,22,26,33], and debridement and trimming and excision of the contused subcutaneous tissues [17,28,30,34] should generally be performed. An appropriate antibiotic protocol and a tetanus immune globulin booster should be administered as well [26,33].

In cases of pantalar dislocation without talar fractures, Boden et al. suggested first reducing the talonavicular joint and then reducing the other joints [28]. Conversely, Bugallo et al. recommend reducing the tibiotalar joint to make other reductions easier [35]. Other authors do not suggest a specific order of reducing. Different case series have reported the use of temporary Schanz pins or K-wires as support for talar manipulation [8].

After achieving complete reduction, it is important to confirm joint stability. The talus typically has good stability in its original position due to the presence of surrounding ligaments and articular surface congruence. Some authors have suggested internal fixation is not necessary unless there is a burst-type open wound or an accompanying fracture [7]. On the other hand, many researchers typically apply one or two K-wires or Schanz pins to increase the articular stability. External fixation is an effective solution for ankle stabilization and also a handy option for the management of daily wound medication [22]. It is unclear, however, which stabilization system is optimal, and the choice is left to the surgeon.

### 4.4. Rehabilitation

Rehabilitation aims to restore the ankle’s range of motion and the patient’s ability to walk. Following surgery or closed reduction, the foot is immobilized and the patient is prevented from putting weight on the injured limb for 6–8 weeks [18,25,26]. Next, physical rehabilitation can begin with active and passive exercises to improve ankle range of motion and muscle strength. Manual techniques and elastic taping can help reduce swelling and soft tissue adhesions. Partial weight-bearing using crutches may be allowed, and the patient may be instructed in gait-training exercises focused on proprioception, balance, and resistance [27]. Before proceeding with full weight-bearing, radiographs should be obtained that are free of the Hawkins sign; MRIs can also confirm no signs of talus AVN [32,43,47].

Four to six months postoperatively, most patients should be able to walk normally without an aid [18,34].

At that point, the rehabilitation protocol should continue to focus on restoring function, such as improving tolerance for walking on uneven ground, climbing and descending stairs, and avoiding obstacles [27]. The goal of rehabilitation is to help the patient return to their previous level of function, including normal daily activities and high-impact recreational activities, despite mild or moderate impairment, within about a year [27,31,36].

Some of the parameters used to assess the recovery of the foot and ankle include the degree of movement in plantar flexion, dorsiflexion, pronation and supination, and specific scores such as the American Orthopaedic Foot and Ankle Society (AOFAS) Ankle-Hindfoot Score [8,20,21,31,37], the Foot Function Index (FFI) and Musculoskeletal Function Assessment (MFA) [28], the Manchester–Oxford Foot Questionnaire-Index (MOXFQ-Index), and the Short-Form Health Survey 36 (sf 36) [30].

### 4.5. Complications

A total closed or open talar dislocation is often a severe injury that could cause a disabled ankle [1]. The severity of the ankle injury and the treatment chosen to reimplant the talus into the mortise are two factors strongly associated with the risk of developing the three main complications: avascular necrosis (AVN), post-traumatic arthrosis [50], and infections.

AVN is the most common long-term complication, estimated to occur in 26% of cases [1], and this percentage may even be underestimated because of the short follow-up of studies [28]. Post-traumatic talar AVN develops after the interruption or compromission of anastomoses from the anterior tibial, posterior tibial, and perforating peroneal arteries that guarantee blood supply network to the entire talar bone [5,51]. For this reason, it is essential to preserve the remaining blood supply to ensure good perfusion of the bone independent of the treatment undertaken [15]. According to several studies, the risk of AVN is higher in addition to vascular compromission when no soft tissues remain attached to the talus [5]. Hosny et al. supported that the onset of AVN may be avoided if the deltoid ligament or the posterior tibial artery that branch off the posterior process are preserved [39]. The risk of post-traumatic talar AVN is over 90% during the first year [52] and remains high until the second-year post injury. It is important to diagnose AVN as soon as possible. However, the Hawkins sign, an indirect sign of vascularization, may only appear on an X-ray 6–8 weeks after the trauma. A positive Hawkins sign shows good vascularization, while a negative sign indicates osteonecrosis. MRI is more effective than X-ray in observing the Hawkins sign, but it is not cost-effective for screening [43,53]. To reduce the risk of AVN, Eda et al. recommend (I) the avoidance of surgery when possible, instead opting for closed reduction to preserve soft tissues and blood supply; (II) starting early rehabilitation; (III) allowing weight bearing after detecting a positive Hawkins sign on X-ray [36].

The second prevalent complication is post-traumatic arthrosis [54]. In the literature, a rate of 22% is reported for radiographic osteoarthritis and a rate of 16% is reported for clinical osteoarthritis [1]. These percentages could be underestimated due to the short follow-ups of the studies in the literature [28]. Arthrosis could involve the tibiotalar, subtalar, or talonavicular or more than one joint together. Regardless, in the absence of talar fracture, it is difficult to predict post-traumatic arthrosis [15].

The third most common complication is infection, associated with poor clinical and functional outcomes. It is correlated with open fractures or extrusions. Nowadays, the incidence of infection has decreased to a range of 25–38%, compared to the past rate of 89%, due to improved antibiotic therapy protocols and soft tissue handling [50,55]. The integrity of the soft tissue works as a barrier against microorganisms and the good blood supply enables the delivery of antibiotics [51]. The timing for reimplanting the talus into the mortise is crucial to prevent infection. Delayed reimplantation increases the infection risk. In cases of total talar extrusion, prompt wound debridement and reimplantation are necessary [55]. These procedures are supported by administering oral antibiotics, and in cases of massive contamination, intravenous antibiotics are used. Primary talectomy and/or arthrodesis are not procedures performed initially; these surgeries are reserved for patients with deep infections [5]. Ely et al. reported a case of below-the-knee amputation after a resistant infection in a patient with open total talar dislocation [56].

Lastly, complex regional pain syndrome type 1 is a rare complication associated with delayed reduction or partial analgesia [57] after direct injury of the tibial nerve.

### 4.6. Limitations

This study has limitations due to the rarity of the condition, leading to unreliable conclusions. An important bias is the presence of case reports as part of the systematic review. Based on our knowledge, there are no randomized controlled trial (RCT) studies in the literature on this topic. A meta-analysis was not carried out because, based on the level of evidence from the studies included in the systematic review, it was not possible to conduct a more in-depth analysis. In addition, the certainty of the evidence was not evaluated. However, there are studies that have not met our inclusion criteria that could be an important resource for future studies on this rare condition. The collected studies were heterogeneous, with varying follow-up timings and durations, making it difficult to assess the long-term outcomes and detect complications. We have also not disclosed the sources of funding for studies included in the review. The quality assessment of the included studies was not carried out according to a standardized method but through the application of a scoring system based on the methodological rigor of the studies examined and the knowledge of the topic.

## 5. Conclusions

Pure total talar dislocation is a rare and complex injury. Early and accurate diagnosis is crucial for effective management to prevent complications such as avascular necrosis, infection, and post-traumatic arthritis. While there is no consensus on the best management approach, principles include early reduction, stabilization with an external fixator for unstable joints, and accurate wound debridement for open dislocations. Rehabilitation is important for restoring function. The continued documentation of cases is essential for refining protocols and improving outcomes. Future studies should focus on developing standardized treatment guidelines for this complex injury.

## Figures and Tables

**Figure 1 jfmk-10-00055-f001:**
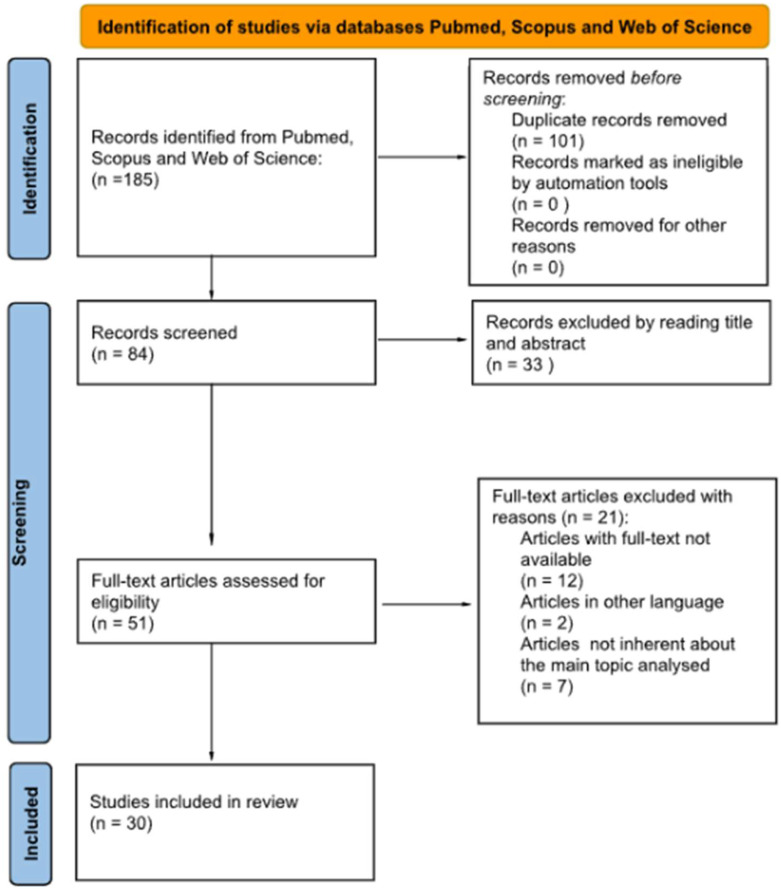
The PRISMA flowchart for the selection and screening of the articles.

**Table 1 jfmk-10-00055-t001:** Summary of the selected studies.

Author and Year	Title	Type of Study	Number of Patients	Open/Closed Dislocation	Reduction	Treatment	Follow-Up (Months)
Ritsema 1988 [13]	Total talar dislocation	Retrospective	5	Closed	Open	-----	48 (36–72)
Papanikolaou 2002 [14]	Successful treatment of total talar dislocation with Closed reduction: A case report	Case report	1	Closed	Closed	Steinmann pin	30
Wagner 2004 [12]	Talar dislocations	Retrospective	6	2 open, 4 closed	Open	Secondary arthrodesis (4), K-wire fixation (1), external fixator (1)	76 (9–204)
Taymaz 2005 [15]	Complete dislocation of the talus unaccompanied by fracture	Case report	1	Closed	Closed	Cast	48
Fleming 2009 [16]	Total talar extrusion: a case report	Case report	1	Open	Open	External fixator followed by cast	12
Apostle 2010 [17]	Reimplantation of a Totally Extruded Talus	Case report	1	Open (missing talus)	-	K-wire fixation	60
Burston 2010 [18]	Open total talus dislocation: clinical and functional outcomes: a case series	Case series + literature review	8	Open (1 missing talus)	Open	Cast (3), external fixator (2), temporary external fixator (1), K-wire + screw fixation (1), screw fixation (1)	42 (13–72)
Vaienti 2011 [19]	Therapeutic management of complicated talar extrusion: Literature review and case report	Case report	1	Open (missing talus)	/	External fixator with antibiotic cement spacer followed by talar reimplantation and arthrodesis	
Gopisankar 2012 [20]	A rare case of Closed pantalar dislocation combined with Lisfranc’s injury—The unusual complex	Case report	1	Closed	Closed	K-wire fixation + cast	12
Gursu 2013 [7]	Closed total dislocation of talus without any accompanying fractures	Case report	1	Closed	Open	Cast	24
Karampinas 2014 [21]	Open talar dislocations without associated fractures	Retrospective	9	Open	Open	Steinmann pins + external fixator	19–23
Breccia 2014 [22]	Treatment and outcome of Open dislocation of the ankle with complete talar extrusion: a case report	Case report	1	Open	Open	External fixator	18
Lee 2014 [23]	Total talar extrusion without soft tissue attachments	Case series	2	Open (missing talus)	-	External fixator with antibiotic cement spacer followed by Ilizarov external fixator and bone graft arthrodesis (1), reimplantation and subtalar fusion with screws (1)	24–96
Kumar 2014 [24]	Closed Talar Dislocation without Associated Fracture a Very Rare Injury, a Case Report	Case report	1	Closed	Closed	K-wire fixation + cast	12
Mohindra 2014 [8]	Early reimplantation for Open total talar extrusion	Retrospective	7	Open	Open	K-wires + external fixator (5), tension band wiring + splint (2)	32 (24–46)
Nanjayan 2014 [25]	Total dislocation of the talus: a case report. Foot Ankle	Case report	1	Closed	Closed	Cast	24
Sié 2014 [26]	Delayed debridement of an Open total talar dislocation reimplanted in the emergency room	Case report	1	Open	Open	Splint	12
Weston 2015 [1]	Systematic Review of Total Dislocation of the Talus	Systematic review	29/86 (no fractures of the talus or adjacent bones)	21 open, 8 closed	27 Open, 2 Closed	Internal fixation (10), K-wire fixation (4), screws (3), pins (1)— external fixation (10), arthrodesis (8)	32
Ranalli 2016 [27]	Rehabilitation Following a Traumatic Dislocation of the Talus: A Case Study	Case study	1	Open	Open	Splint	6
Boden 2017 [28]	Complications and Functional Outcomes After Pantalar Dislocation	Case report	19	14 open, 5 closed	17 open, 2 closed	External fixator (9), K-wire fixation (4), K-wires + external fixator (3), transtibial amputation (1)	41 (3–157)
Kwak 2017 [29]	Six-year survival of reimplanted talus after isolated total talar extrusion: a case report	Case report	1	Open	Open	K-wire fixation + external fixator	72
Ruatti 2017 [30]	Replacement after Talar Extrusion	Case report	1	Open	Open	Antibiotic cement spacer followed by total ankle prosthesis	24
Steinmetz 2017 [31]	Isolated Open Total Talar Dislocation: A Case Report and Literature Review	Case report	1	Open	Open	External fixator	18
Yapici 2019 [32]	Open reduction of a total talar dislocation: A case report and review of the literature	Case report + literature review	1	Closed	Open	K-wire fixation + cast	36
Genena 2020 [33]	A Case Report of an Open Pan-Talar Dislocation	Case report	1	Open	Open	Steinmann pin	12
AlMaeen 2020 [34]	Complete Revascularization of Reimplanted Talus After Isolated Total Talar Extrusion: A Case Report	Case report	1	Open	Open	Cast	12
Bugallo 2021 [35]	Closed posteromedial total talus and fibula dislocation without fracture.	Case report	1	Closed	Open	Cast	2
Eda 2021 [36]	Closed total talar dislocation without fracture in a rare college athlete case	Case report	1	Closed	Closed	Cast	18
Khan 2022 [6]	Pantalar dislocation: a rare presentation with review of treatment method	Case report + literature review	1	Closed	Open	K-wire fixation + splint	6
Leonetti 2023 [37]	Total Talar Prosthesis, Learning from Experience, Two Reports of Total Talar Prosthesis after Talar Extrusion and Literature Review	Case series + literature review	2	Open (missing talus)	-	External fixator with antibiotic cement spacer followed by reimplantation and arthrodesis, total talar replacement and final triple arthrodesis (1), external fixator with antibiotic cement spacer followed total talar replacement and final total ankle prosthesis (1)	60

## Data Availability

Not applicable.

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
