# Peer review of "Pantalar Intact Dislocation: A Systematic Review"

_jfmk, 2025, doi:10.3390/jfmk10010055_

Round 1
Reviewer 1 Report (New Reviewer)
Comments and Suggestions for Authors
A very interesting study but have some comments.
I think the manuscript can be accepted with only minor revisions.
I think this manuscript is very interesting and the author has carried out some studies in this area.
1- you should indicate the trade mark ™ of the software used
2- you should revise the way you write (line 136 and 140, I've identified some notes in the document)
3- Table 1 should be subtitled, and it should be reorganized. as it is a very long table I suggest that it is organized by chronological date or by type of injury.
4- There are some parts of sentences written in italics, but I don't understand why write in itlaic them?
5- When you write some acronyms for the first time, you should name them and only then use the acronyms (e.g. CT and MRI).
I attach some notes on the manuscript.
Very good work! Congratulations.

Author Response
|
Response to Reviewer 1 Comments
|
||
|
Summary |
|
|
|
Thank you very much for taking the time to review this manuscript. Please find the detailed responses below and the corresponding revisions/corrections highlighted in the re-submitted files. |
||
|
Comments 1: you should indicate the trade mark ™ of the software used Response 1: Thank you for your comment. No software was used, the PRISMA flowchart was created by us authors.
|
||
Comment 2: you should revise the way you write (line 136 and 140, I've identified some notes in the document)
Response 2: Thank you for your comment. We have revised the way we write (line 136 and 140) we have followed your notes in the manuscript.
Comment 3: Table 1 should be subtitled, and it should be reorganized. as it is a very long table I suggest that it is organized by chronological date or by type of injury.
Response 3: Thank you for your comment. We have organized table 1 by chronological date as you suggested.
Comment 4: There are some parts of sentences written in italics, but I don't understand why write in itlaic them?
Response 4: Thank you for your comment. We have changed the italics.
Comment 5: When you write some acronyms for the first time, you should name them and only then use the acronyms (e.g. CT and MRI).
Response 5: Thank you for your comment. We have written the full name of the acronyms.
We have followed and corrected all the notes you attached in the manuscript.
We thank you for your valuable work and for the congratulations.

Reviewer 2 Report (New Reviewer)
Comments and Suggestions for Authors
Aim of this Review was to systematically examine and analyze the available literature on pure total talar dislocation, focusing on its epidemiology, clinical presentation, imaging techniques, surgical options, rehabilitation protocols, and complications.
Although "conclusions" section has been written in a very scientific way, emphasizing various clinical-instrumental aspects of pure total talar dislocation, with useful ideas for the clinical management of these patients, I believe some minimal changes to the text are useful:
1) "introduction" section: I improve lines 44-53 because I believe there are too many sentences on the "history" (even ancient) of pure total talar dislocation
2) "materials and methods" section: better explain the inclusion and exclusion criteria in the text
3) "results" section: improve the captions of Figures/Tables to make them better understood
4) improve the quality of English
Comments on the Quality of English LanguageMinor revision of English quality
Author Response
|
Response to Reviewer 2 Comments
|
||
|
Summary |
|
|
|
Thank you very much for taking the time to review this manuscript. Please find the detailed responses below and the corresponding revisions/corrections highlighted in the re-submitted files.
|
||
Comment 1: introduction" section: I improve lines 44-53 because I believe there are too many sentences on the "history" (even ancient) of pure total talar dislocation.
Response 1: Thank you for your comment. We improved the rich 44-53, removing many sentences on the "history".
Comment 2: "materials and methods" section: better explain the inclusion and exclusion criteria in the text.
Response 2: Thank you for your comment. We better explain the inclusion and exclusion criteria in results section.
Comment 3: "results" section: improve the captions of Figures/Tables to make them better understood.
Response 3: Thank you for your comment. We have improved the captions of Figures/Tables.
Comment 4: improve the quality of English
Response 4: Thank you for your comment. We have improved the quality of English. In “Non-published Material” you can find the certificate of editing.

Reviewer 3 Report (New Reviewer)
Comments and Suggestions for Authors
Why is the search until june 2023?
Would you mind attaching the search strategy, e.g., in appendixes?
Please extend the description of the methods:
- who identified the research (+ resolved conflicts) at the level of titles, abstracts and full texts
- who extracted the data
- how the credibility of the research was assessed (how many people did it, who resolved conflicts) - the Jadad scale does not contain important information about unconcealed allocation; why have you decided to use the Oxford scale? you did not include it in the protocol
- how the results were synthesized.
Prisma must be corrected:
- 'records after duplicated removed' means how many records remained after removing, not how many have been deleted
- the description looks pretty vague looking at the numbers in Prisma
- please list the reasons for exclusion of the individual study (what are the exclusion criteria?)
You did not report on the sources of funding for the studies included in the review
Author Response
|
Response to Reviewer 3 Comments
|
||
|
Summary |
|
|
|
Thank you very much for taking the time to review this manuscript. Please find the detailed responses below and the corresponding revisions/corrections highlighted in the re-submitted files.
|
||
Comment 1: Why is the search until june 2023?
Response 1: Thank you for your comment. Because we explored the main electronic databases (Pubmed, Web of Science, Scopus) in September 2023.
Comment 2: Would you mind attaching the search strategy, e.g., in appendixes?
Response 2: Thank you for your comment. We have inserted the search strategy in the appendix, as you suggested.
Methods issues:
Comment 3: Who identified the research (+ resolved conflicts) at the level of titles, abstracts and full texts.
Response 3: Thank you for your comment. M.D.C, E.D and M.A.M identified the research at the level of titles, abstracts, and full texts.
Comment 4: Who extracted the data.
Response 4: Thank you for your comment. S.P and A.S extracted the data.
Comment 5: How the credibility of the research was assessed (how many people did it, who resolved conflicts) - the Jadad scale does not contain important information about unconcealed allocation; why have you decided to use the Oxford scale? you did not include it in the protocol.
Response 5: Thank you for your comment. All authors were involved in defining the credibility of the research. We used the Oxford scale, which we consider more appropriate. Also, as specified in the study limitations, an important bias is the presence of case reports as part of the systematic review. Based on our knowledge, there were no randomized controlled trial (RCT) studies in the literature on the topic.
Comment 6: How the results were synthesized.
Response 6: Thank you for your comment. In the text you can find the various sub-sections of the results well summarized.
Prisma issues:
Comment 7: 'Records after duplicated removed' means how many records remained after removing, not how many have been deleted.
Response 7: Thank you for your comment. We attach the link from which we took the PRISMA flowchart template that we have always used in our systematic reviews. https://www.prisma-statement.org/prisma-2020-flow-diagram
In our opinion, this is the correct format for a PRIMSA flowchart. If you know another PRIMSA flowchart template, could you attach it.
Comment 8: The description looks pretty vague looking at the numbers in Prisma.
Response 8: Thank you for your comment. In the text you will find a full description of the numbers in PRISMA.
Comment 9: please list the reasons for exclusion of the individual study (what are the exclusion criteria?)
Response 9: Thank you for your comment. They have been carefully described by line 95 to 159. If you need further clarification we can provide.
Comment 10: You did not report on the sources of funding for the studies included in the review.
Response 10: We have included this important comment in the limitations of the study. Thanks.

This manuscript is a resubmission of an earlier submission. The following is a list of the peer review reports and author responses from that submission.
Round 1
Reviewer 1 Report
Comments and Suggestions for Authors
Firstly, the studies collected for this research are highly heterogeneous, with varying degrees of soft tissue injury severity, treatment methods, and follow-up durations. Additionally, the final sample included 30 articles, most of which were case reports. During the initial screening, 12 articles were excluded due to lack of full-text access, which might have influenced the proportion and potentially led to different conclusions. As a result, this study could not consolidate a definitive treatment guideline.
Secondly, the review article does not present any new information or recommendations.
Author Response
Thank you for your valuable contribution to the manuscript review process.
As repeatedly stated in the manuscript's text, this is a rare affection for which the data present today in the literature do not allow us to draw universally agreed guidelines.
Continued documentation of cases is essential for refining protocols and improving outcomes.
Reviewer 2 Report
Comments and Suggestions for Authors
Thank you for the opportunity to review this manuscript. I have a few recommendations and questions to enhance the clarity and depth of the study:
- Placement of Methodology Information:
- The section stating, “In the identification phase, a total of 185 articles were found. Out of these, 101 studies were excluded because they were reported more than once. After the first preliminary screening, 33 articles were removed because their titles or abstracts were not consistent with the main topic. Further 21 studies were excluded: 12 because the full text was not available, 2 because they were not written in English and 7 because they were not related to the main topic analyzed. Therefore 30 studies were included at the final check. All authors collectively re-evaluated their choices to ensure accuracy, and the PRISMA flow-chart is depicted, and the selected articles are summarized according to inclusion and exclusion criteria (Figure 1).”
- This content and Figure 1 should be moved to the Results section. In the Methods section, the authors should focus on describing the inclusion and exclusion criteria in detail, following the PRISMA guidelines.
- Quality Assessment Methodology:
- The authors state that “Five authors (M.D.C, E.D, M.A.M, S.P, and A.S) individually read the selected articles, evaluating and discussing their quality.”
- Please describe the method used for quality assessment of the included studies. Was a standardized tool or scoring system used to evaluate study quality? If not, this should be mentioned as a limitation.
- Addressing Bias:
- The included studies range from case reports to reviews. How did the authors address potential biases inherent in case reports when synthesizing the findings? A discussion on this would strengthen the manuscript.
- Recommendations for Future Studies:
- Can the authors provide specific recommendations for improving reporting standards in future studies on pure total talar dislocation?
- Further Analysis:
- Did the authors consider performing additional analyses, such as a meta-analysis or sensitivity analysis, where possible? If not feasible, please include an explanation in the manuscript.
- Risk of Bias Assessment:
- Did the authors evaluate the risk of bias for the included studies? If not, this should be addressed as a limitation.
- Certainty of Evidence:
- Was the certainty of evidence assessed using a framework such as the GRADE system? If not, consider mentioning it as a limitation.
- Search Details:
- Please include detailed results of the database searches (e.g., search terms, combinations of keywords, and the exact date of the search) in Appendix 1. This will ensure replicability.
- Reflection on Limitations:
- If the manuscript does not cover the answers to the above questions, I recommend adding these points to the Limitations section to provide a more comprehensive discussion of the study's constraints.
Author Response
Author's Reply to the Review Report (Reviewer 2)
Thank you for your valuable contribution to the manuscript review process.
Below you will find the revision point by point and the changes in the text marked in red.
Comment 1: Placement of Methodology Information: The section stating, “In the identification phase, a total of 185 articles were found. Out of these, 101 studies were excluded because they were reported more than once. After the first preliminary screening, 33 articles were removed because their titles or abstracts were not consistent with the main topic. Further 21 studies were excluded: 12 because the full text was not available, 2 because they were not written in English, and 7 because they were not related to the main topic analyzed. Therefore 30 studies were included at the final check. All authors collectively re-evaluated their choices to ensure accuracy, and the PRISMA flow-chart is depicted, and the selected articles are summarized according to inclusion and exclusion criteria (Figure 1).”
This content and Figure 1 should be moved to the Results section. In the Methods section, the authors should focus on describing the inclusion and exclusion criteria in detail, following the PRISMA guidelines.
Response 1: The contents and figure 1 have been moved to the results section as correctly requested.
Comment 2: Quality Assessment Methodology: The authors state that “Five authors (M.D.C, E.D, M.A.M, S.P, and A.S) individually read the selected articles, evaluating and discussing their quality.”
Please describe the method used for the quality assessment of the included studies. Was a standardized tool or scoring system used to evaluate study quality? If not, this should be mentioned as a limitation.
Response 2: Unfortunately, we did not find a standardized method of quality assessment. However, we applied our scoring system based on the rigor of the studies examined and the knowledge of the topic. As requested, we are mentioning this aspect in the limitations of the study
Comment 3: Addressing Bias: The included studies range from case reports to reviews. How did the authors address potential biases inherent in case reports when synthesizing the findings? A discussion on this would strengthen the manuscript.
Response 3: Thanks for your suggestion, we have added this important bias as part of the discussion in the limitations section.
Comment 4: Recommendations for Future Studies: Can the authors provide specific recommendations for improving reporting standards in future studies on pure total talar dislocation?
Response 4: Thank you for your comment, we have included recommendations for future studies in the conclusions
Comment 5: Further Analysis: Did the authors consider performing additional analyses, such as a meta-analysis or sensitivity analysis, where possible? If not feasible, please include an explanation in the manuscript.
Response 5 We did not perform a meta-analysis because based on the level of evidence of the studies included in the systematic review it was not possible to include a more in-depth analysis.
This explanation has been inserted in the text
Comment 6: Risk of Bias Assessment: Did the authors evaluate the risk of bias for the included studies? If not, this should be addressed as a limitation.
Response 6: Based on your suggestions, the risk of bias has been included in the limitations as requested.
Comment 7: Certainty of Evidence: Was the certainty of evidence assessed using a framework such as the GRADE system? If not, consider mentioning it as a limitation.
Response 7: Unfortunately, the certainty of evidence was not evaluated and therefore was included as a limitation.
Comment 8: Search Details: Please include detailed results of the database searches (e.g., search terms, combinations of keywords, and the exact date of the search) in Appendix 1. This will ensure replicability.
Response 8: Thank you for your comment, research details have been included.
Comment 9: Reflection on Limitations: If the manuscript does not cover the answers to the above questions, I recommend adding these points to the Limitations section to provide a more comprehensive discussion of the study's constraints.
Response 9: Thank you for your comment. We have added these points to the limitations section to provide a more comprehensive discussion of the study's constraints.
Reviewer 3 Report
Comments and Suggestions for Authors
In medicine, there is not only a lot of new territory worth exploring at the molecular level, there are also many questions at the macroscopic level in the "old" medical arts. Therefore, an important achievement of this work is to point out an "orchid" of surgery and to survey the associated state of the art. In view of the (for me surprisingly high, given the low incidence of trauma described above) number of 30 sources used, every reference is important. Therefore, beyond the economic "war" of the large publishing houses, I am surprised that possibly existing publications outside the not necessarily objective preliminary evaluation by Clavariate & Co. were not even searched for - the authors still make their own evaluation and thus provide quality control by filtering out. Since Ilizarov at the latest, we have also known in traumatology that looking over the fence can be more than worthwhile. I would like the authors to be sure that, beyond Pubmed, Scopus and WoS, there are no significant publications on this topic that can be found with our resources. With the review on an exotic topic, a reference is created that will foreseeably not be revised for a long time, after which the topic appears to have been dealt with.
The two foreign-language publications found seem therefore worth analyzing in more detail. Even Google Scholar provides me with 14 pages of references, which can easily be compared with the previous findings. Libraries offer a surprisingly high number and quality of search tools, seeking for literature from pre-digital times. In the much-maligned MDPI, now ignored by Clavariate, I immediately find a publication by the neighbors from Padua. In science, we should also look at what ended up in the poison cabinet of the (here Anglo-American) Inquisition. After all, the authors themselves submit to MDPI. May be, the authors caught all relevant publications, but the fishing net seems to me not to be wide enough to really trust in this believe.
Terminologia anatomica is the "ISO standard" for medicine. Therefore, the anatomical term should be mentioned first (A. tibialis dors., in italics - difficult to implement with the MDPI format templates, I know), followed by the scientifically established colloquial English term, which is then used in the following. Since the royal road of science is the question, I would like authors who are as familiar as possible with the problem area to name the most urgent questions that need to be answered in order to make progress in the treatment of talar dislocation. Perhaps even hypotheses can be formulated that can (also) be tested by others, as biomechanics has long been a close friend of trauma surgery. If there is already a lack of technically substantiated observations on biomechanics, why the desire to publish in a journal that (fortunately) focuses on functional morphology and kinesiology, both of which are related to biomechanics? I believe that this interesting article would be better suited to the target group and more appropriate in terms of content in a journal on trauma surgery.
Small formal hints:
You can save a lot of space by formatting the column width of the table (whose heading is missing) appropriately. If the format template resists: insert the abbreviation # and FA in the heading and use it in the table, the columns with the numbers will be very narrow.
The bibliography program has written the abbreviated Italian month names in front of the year numbers for many sources; this should be corrected. The sources [6] and [34] are incomplete.
Lines 173 & 174: The structures should be named anatomically correct.
Author Response
Author's Reply to the Review Report (Reviewer 3)
First of all, thank you for the surprisingly detailed description of your review. Your comments are typical of a colleague who cultivates the medical art with great enthusiasm, and this can only enrich our manuscript. Collaborating with a passionate colleague is always a reason for human and professional growth.
Our research was very accurate, and the main electronic databases explored have provided us with a large number of articles but on closer analysis they were selected 30, which for us represents a number much higher than our expectations.
The anatomical term was mentioned in italics as required. The choice of the journal, as you correctly observed, stems from the desire to give space to growing journal that offer a wide range of benefits for young researchers who want to explore the medical art.
Small formal hints:
Comment 1: You can save a lot of space by formatting the column width of the table (whose heading is missing) appropriately. If the format template resists: insert the abbreviation # and FA in the heading and use it in the table, the columns with the numbers will be very narrow. Response 1: We have formatted the table according to your request.
Comment 2: The bibliography program has written the abbreviated Italian month names in front of the year numbers for many sources; this should be corrected. The sources [6] and [34] are incomplete. Response 2: We have revised the references according to your comment.
Comment 3: Lines 173 & 174: The structures should be named anatomically correct. Response 3: Thank you for your important suggestion, we have corrected it.
Round 2
Reviewer 2 Report
Comments and Suggestions for Authors
Thank you for addressing my inquiries. I have received satisfactory answers to all my questions and have no further concerns at this time.
Reviewer 3 Report
Comments and Suggestions for Authors
the authors changed the formal things, and it's content is intersting, but this does not change the fact the manuscript contains no grain of Function Morphology or Kinesiology, and even does not cover the scope of the journal:
- Exercise and Physical Health
- Sports Psychology and Cognitive Functioning
- Muscle Structure and Musculoskeletal Disorders
- Anatomy and Kinesiology
- Adapted Physical Activity for Health Promotion
- Rehabilitation and Rheumatology
- Sports Medicine, Injury Prevention and Treatment
- Strength and Power
- Nutrition and Body Composition
- Physical Activity and Neurodegeneration
- Postural Control and Balance
- Resistance Training
- Sport Physiology and Performance
- Athlete Monitoring and Management
- Team Sports and Technology